# Evolution and Classification of Musaceae Based on Male Floral Morphology

**DOI:** 10.3390/plants12081602

**Published:** 2023-04-10

**Authors:** Wandee Inta, Paweena Traiperm, Saroj Ruchisansakun, Steven B. Janssens, Unchera Viboonjun, Sasivimon C. Swangpol

**Affiliations:** 1Department of Plant Science, Faculty of Science, Mahidol University, Bangkok 10400, Thailand; 2Meise Botanic Garden, Nieuwelaan 38, 1860 Meise, Belgium; 3Department of Biology, KU Leuven, 3001 Leuven, Belgium; 4Leuven Plant Institute, KU Leuven, 3001 Leuven, Belgium

**Keywords:** Bayesian inference, floral evolution, infrageneric classification, morphometric analysis, phenetics

## Abstract

Classification of the banana family (Musaceae) into three genera, *Musa*, *Ensete* and *Musella*, and infrageneric ranking are still ambiguous. Within the genus *Musa*, five formerly separated sections were recently merged into sections *Musa* and *Callimusa* based on seed morphology, molecular data and chromosome numbers. Nevertheless, other key morphological characters of the genera, sections, and species have not been clearly defined. This research aims to investigate male floral morphology, classify members of the banana family based on overall similarity of morphological traits using 59 banana accessions of 21 taxa and make inferences of the evolutionary relationships of 57 taxa based on ITS, *trnL-F*, *rps16* and *atpB-rbcL* sequences from 67 Genbank and 10 newly collected banana accessions. Fifteen quantitative characters were examined using principal component analysis and canonical discriminant analysis and 22 qualitative characters were analyzed by the Unweighted Pair Group Method with an Arithmetic Mean (UPGMA). The results showed that fused tepal morphology, median inner tepal shape and length of style supported the three clades of *Musa*, *Ensete* and *Musella*, while shapes of median inner tepal and stigma classified the two *Musa* sections. In conclusion, a combination of morphological characters of male flowers and molecular phylogenetics well support the taxonomic arrangement within the banana family and the *Musa* genus and assist in selection of characters to construct an identification key of Musaceae.

## 1. Introduction

*Musa* L., *Ensete* Bruce ex Horan. and *Musella* (Franch.) C.Y. Wu ex H.W.Li belong to the banana family [1]. In 1947, Cheesman separated *Ensete* from *Musa* based on a monocarpic habit and divided the genus *Musa* into four sections mainly on inflorescence orientation, seed shapes and chromosome numbers, including sect. *Musa* (inflorescence pendant, seed sub-globose, compressed or irregular angulate and *x* = 11), sect. *Rhodochlamys* (inflorescence erect, seed sub-globose, compressed or irregular angulate, *x* = 11), sect. *Australimusa* (inflorescence pendant, seed sub-globose, *x* = 10) and sect. *Callimusa* (inflorescence erect, seed cylindrical, barrel-shaped, or top-shaped, *x* = 10) [2]. Later, Argent placed *M. ingens* into a new sect. *Ingentimusa* (inflorescence erect, seed sub-globose to irregular angulate, *x* = 7) [3]. Based on molecular phylogenetic analyses and chromosome numbers, Häkkinen later proposed a new sectional classification of the genus *Musa* by merging the five sections into two: *Musa* sect. *Musa* (*n* = *x* = 11) and *Musa* sect. *Callimusa* (*n* = *x* = 10, 9, 7) [4], nevertheless, did not refer to the morphologically discriminating characters between the two sections. In 2015, Swangpol and her team reported a new species of *Musa*, *M. nanensis* with remarkable staminal features and a unique arrangement of tepals [5]. The species possesses six stamens fused at the base instead of five isolated stamens. In addition, three outer tepals are fused with three inner ones instead of a compound tepal of fused three outer and two inner ones with a free tepal as in other members of Musaceae. Here, we present a detailed comparative study on the floral structures of species within the Musaceae.

Though Musaceae is a monophyletic taxon based on nuclear and chloroplast genetic analyses [6,7,8], it has diverse floral morphological characters especially the perianth and androecium [1,9,10]. *Musa nanensis* is the only taxa in the order Zingiberales with actinomorphic flowers and its phylogenetic position in the family is unknown. To improve the classification within the banana family, a molecular phylogenetic analysis of the Musaceae (incl. *M. nanensis*) is conducted and used to investigate the floral morphological evolution of wild bananas. In addition, keys to the genera, sections and species based on male floral characters were generated.

## 2. Results

### 2.1. Phylogenetic Analysis

The Bayesian phylogenetic reconstruction from combined dataset including nrDNA, trnL-F spacer, rps16 intron and atpB-rbcL spacer (Appendix A) shows monophyletic group of Musaceae placing at the base of the Zingiberales. Musella was grouped with Ensete and separated from Musa. The relationship within Musa shows two separated clades, which consist of the Musa section (incl. Rhodochlamys) and the Callimusa section (incl. Australimusa). Musa nanensis was positioned in the Musa section.

### 2.2. Multivariate Analysis

Principal Component Analysis (PCA) for 15 quantitative characters of floral morphology from 59 OTUs of *Musa*, *Ensete* and *Musella* determined first four principal components with eigenvalues greater than 1, which provided 80.51% of all observed variation (Table 1). The high values of component matrix from 0.700 to 0.999 indicated the correlation to factor analysis (Table 2). The first principal component provided 35.26% of the variation and was strongly positively correlated with width of compound tepal, width of lateral outer compound tepal lobe, length of lateral outer compound tepal lobe, width and length of central outer compound tepal lobe and length of central outer compound tepal lobe (Table 2). The second principal component provided 26.53% of the variation and shows a strong positive correlation with length of median inner tepal and length of style base to stigma head and high negative correlated with length of inner compound tepal lobe (Table 2).

The scatter plot of the first two principal components from 15 floral morphological measurements indicated that all accessions clustered into two separated groups of Musaceae, i.e., *Ensete* taxa separated from those of *Musa* and *Musella* (Figure 1). However, PCA based on quantitative analysis of male flowers cannot distinguish *Musella* from *Musa*. The first and second principal components strongly distinguished *Ensete* from *Musa* and *Musella*, and the first principal component weakly separated *Musella* from *Musa* (Table 2).

Canonical Discriminant Analysis (CDA) provided 100% of the variation between the three genera in the first two dimensions (Table 3, Figure 2). The results of the CDA indicated that the characters most significantly separated flowers of *Ensete* from *Musa* and *Musella* were, i.e., length from style base to stigma head, width of ovary, length of median inner tepal, length of filament and length of ovary (Table 4—Function 1). The characters that separated *Musella* from *Ensete* and *Musa* includes length of compound tepal, length of anther, width of lateral outer compound tepal lobe, width of compound tepal, length of inner compound tepal lobe, width of central outer compound tepal lobe, width of inner compound tepal lobe, length of central outer compound tepal lobe and width of median inner tepal (Table 4—Function 2).

### 2.3. Morphological Analysis

The cluster analysis was computed based on UPGMA of 22 characters (Figure 3, Figure 4 and Figure 5) from 59 samples of the Musaceae male flowers and presented in Figure 6. *Ensete* is grouped with *Musella* and separated from *Musa*. The four species of *Ensete* are clearly separated from *Musella*. There are two groups in the *Musa* cluster, the first includes *Musa* (incl. *Rhodochlamys*) and the second is *Callimusa* (incl. *Australimusa* & *Ingentimusa*).

The dissimilarities of the male floral morphological characters between the banana species were used in constructing a key to species of the two sections within *Musa*.

### 2.4. Morphological Characters in the Classification of Musaceae

The phylogeny shows that bilateral perianth with five-fused compound tepals and one free median inner tepal is commonly founded within Musaceae, while radial perianths with six-fused tepals is the autapomorphic character found in *M. nanensis* (Figure 3B(b), Figure 4B(e) and Figure 7). In Zingiberales, bilateral perianths are mostly found excepted the asymmetric flowers of Heliconiaceae (Figure 7). The zygomorphic flower is ancestral state in this order.

Based on the shape of the outer tepal-lobe apex and inner tepals, there are two types of Musaceae flowers (Figure 3C). Shared character is the flowers with the acute outer tepal-lobe apex and ovate inner tepals, which were found in *Musa* and *Musella*. On the other hand, shifting to the flower with round outer tepal-lobe apex and long acicular inner tepals were only founded in *Ensete* (Figure 8A).

All Musaceae members, except for *M. nanensis*, have free median inner tepal (Figure 8B). Although the shape of this tepal is various, especially in *Musa*, it is always tricuspidate in *Ensete*. The obovate median inner tepal found in *Musella* is also found in *Musa textilis*. In the section *Callimusa*, oblong median tepal is ancestral state, shifting to lanceolate twice and from lanceolate to obovate once. In the section *Musa* (including *Rhodochlamys*) the elliptic median tepal shape is the ancestral state, shifted to ovate three times, and then changed to the median tepal that fused with other tepals (Figure 4A and Figure 8B). The ancestor of median inner tepal shape cannot be inferred.

Three stigma shapes were found in the male flowers of Musaceae (Figure 5B). Clavate stigma, found in the section *Musa* and in *Ensete* species, is a plesiomorphic character of the family. A spatulate stigma positioned on a slender style is a synapomorphy of *Callimusa* (incl. *Australimusa*). A capitate stigma is an autapomorphic character in *Musella* (Figure 9A). Our results indicate that a clavate stigma is the ancestral state within Musaceae.

The style length is various among the different banana groups (Figure 5D). The style and stigma of *Ensete* are reduced in male flowers with a style length shorter than 18 mm, being shorter than the length of the filament and a stigma length that is less than 0.5 mm. The style of *Musella* and *Musa* are as long as the length of the compound tepal except for some cultivated *M. ornata* where it is very short (Figure 9B and Appendix A). A long style with a stigma that is positioned at the same height as the compound tepal is the ancestral state within Musaceae.

Differences observed along male floral morphological characters in relation to the phylogenetic hypothesis (Figure 7, Figure 8 and Figure 9) were used in to construct a key for the classification at inter- and intraspecific level (see all keys in Section 3)

## 3. Discussion

Previous study on overall morphology and characters of perianths by Simmonds [30] have paved way towards the clarification of the classification of Musaceae, however, could not solve generic status clearly. Recently, works on stem, inflorescence, seed morphology and molecular biology of the family independently created debates on taxonomic arrangement of the three genera members [6,7,31,32]. Our investigation on the male floral morphology concurrently with molecular study, however, initiate a set of discriminated morphological characters based on phylogenetic inference and able to resolve the infrafamilial evolutionary relationship within Musaceae.

Analyses of the male floral morphology accompanied by the molecular data in our study confirmed that the characters can be used to classify Musaceae at generic level into *Musa*, *Ensete*, and *Musella* (Appendix A) as well as to construct an identification key (Figure 7, Figure 8 and Figure 9).

*Ensete* possesses three outer tepals which fuse only at the base and adnate at the margin upwardly. The genus is separated from the other genera by round outer tepal apex lobes with long acicular inner tepals and tricuspidate median inner tepal. These characters were also used in a general key to describe this genus (Figure 3, Figure 4 and Figure 5). Meanwhile, a monocarpic habit is not a suitable character to discriminate at genus level, since *Ensete* plants can produce suckers in cultivation or under abiotic stress [31].

*Musella* is the sister genus of *Ensete* based on molecular phylogenetic evidence [6,7], in contrast to morphological indications including rhizomatous habit and perianth is similar to *Musa* [26,30]. Meanwhile, similarities in inflorescences were used to group *Musella* with *Ensete* [2]. However, our investigation fortified that they are separated genera by their genetics, which is in contrast to the conclusions of Simmond [6] and Bakker [7], but supports the results of Christelova et al. [32] and Janssens et al. [8]. Moreover, the molecular phylogeny indicated that *Ensete* and *Musella* share their ancestor.

Our results agreed with work of Christelova [32] that the capitate stigma (Figure 5A) of *Musella* separates it from the other genera, therefore, this character has a high discriminative power and can be easily used in an identification key. The generic classification of *Musella* remains unclear if only morphological characters were used, while the combination of morphology and molecular phylogenetics can clearly resolve the conflict of these classifications. *Musella* was placed under *Musa* based on a polycarpic habit and the perianth structures [26,30]. However, several characters of *Musella*, i.e., dwarf, congested pseudostems, compact rosette inflorescences and embryological characters, placed *Musella* as a separated genus [33,34,35,36]. The molecular data suggested that *Musella* is closely related to *Ensete* [6,7]; the two genera possess quite similar inflorescences [2,37]. However, the male floral characters of *Ensete* and *Musella* are obviously different. The two genera are grouped in the same cluster.

In spite of these evidences, it was found that, judging from molecular phylogenetics combined with multivariate analyses of the male flowers and the degree of perianth fusion, *Musella* is more similar to *Musa* than to *Ensete*. The fact indicated that the perianth fused at base of *Ensete* are apomorphic in Musaceae. Furthermore, the shared characters of having a perianth fused until the base of the apex in *Musella* and *Musa* suggests that this character is more primitive than a perianth fused at base. Moreover, the long acicular lateral inner tepal, tricuspidate free tepal and reduced style and stigma in male flowers can satisfactorily be used to separate *Ensete* from *Musa* and *Musella*. The tricuspidate shape of the free tepal was often found in *Ensete* [2], the obovate shape in *Musella* while the elliptic, ovate, oblong, lanceolate and obovate shapes in *Musa*. *Musa nanensis* is the only taxon which has a median tepal that is fused with the perianth tube [5].

The multivariate analysis of the 15 male floral morphological characters did not separate the genus *Musa* at sectional level, whereas the molecular phylogeny clearly divided the genus into two sections: *Musa* + *Rhodochlamys* and *Callimusa* + *Australimusa* [6,7,30].

Discriminating characters that delimit the genera of Musaceae are shapes of median inner tepals and stigmas. The median inner tepals of *Musa* species which varies in shapes and length and apical shapes and length clearly separated the genus into two groups. These variations in floral morphology, especially in perianth and androecium dissimilarities, which also occur in the other Zingiberales taxa, may have caused by differences in pollination syndrome [4,9,10]. Moreover, pollinator syndromes in Musaceae are assumed to have switched several times between bats and birds or other animals, except for *Musella*, which is always pollinated by insects [38,39]. Banana species with erect buds, excluding *Musella*, are suitable to be pollinated by sunbirds, whereas those with pendent buds are suitable to be pollination by bats [38,40]. Watery and gelatinous nectar types attract different pollinators and were suggested to be related to bud and floral positions. Watery nectar is found in erect inflorescences, whereas gelatinous nectar is found in horizontal (later pendent) inflorescences [41]. In the genus *Musa*, the median inner tepal changes in length and shape and long free median inner tepals and perianth tubes were found in erect inflorescences, except for *M. rubra* and *M. siamensis*. Conversely, short free median inner tepals were found in pendent inflorescences. Additionally, bats can use their tongues to lick up the sticky nectar from pendent inflorescence, whereas birds can take nectar from both erect and pendent inflorescences [41]. Therefore, fluidity of the nectar depends likely corroborate floral and bud orientations.

Our results indicated that, in bananas, shapes of the median inner tepal opposing nectary gland is different in erect versus pendent inflorescences are different. The elliptic median inner tepal shape with boat-shaped curve easing logging of sticky nectar is found in *Musa* and *Ensete* with hanging inflorescences and pollinated by long-tongued fruit bats. On the other hand, elliptic shape is found in some species of the section *Musa* and the former section *Rhodochlamys*, i.e., *M. rubra* and *M. siamensis*, with erect inflorescences pollinated by birds [40,42]. The difference between the pollinators may also be related to changing length of free median inner tepal, floral direction and viscosity of nectar in the section *Musa*. Moreover, some species of *Rhodochlamys*, i.e., *M. ornata*, *M. velutina* and *Callimusa* + *Australimusa*, do not possess boat-shaped curve of median inner tepals, the tepal length is as long as that of the fused tepal and form perianth tube which contains higher volume of watery nectar [43]. Though both *Rhodochlamys* and *Callimusa* + *Australimusa* were pollinated by birds and have long median inner tepal, the median inner tepal shape is different. The increasing of floral size of bananas in erect inflorescences (Appendix A) suggested adaptation to produce large amount of nectar for bird pollinators [43].

The pistil size of the banana flowers in the genus *Musa* is reduced in male stage [44]. Two shapes of stigma, clavate and spatulate, were found, in the section *Musa* and in *Callimusa* + *Australimusa*, respectively. The spatulate stigma can be distinguished from the clavate one by its slender style and clearly separation of stigma head at connective zone. The stigma shapes of the two sections are unrelated to the position of the inflorescences [6,7,30,45,46,47].

Curiously, *Musa nanensis* found recently by Swangpol and her team in Thailand is unique in floral symmetry and its placement within the *Musa* section have been in question [5]. The molecular phylogenetics of combined ITS, *trnL-F*, *rps16* and *atpB-rbcL* sequences did not distinguish *M. nanensis* from Musaceae, on the other hand, placed it closely to the section *Musa* + *Rhodochlamys* with the erect inflorescence, the tubular flower and the long free tepal. The clavate stigma of *M. nanensis* is also a key character to classify it within the *Musa* section. Finally, within the section, though *M. nanensis* is similar to *M. rubra* (often called previously by its synonym, *M. laterita* [48]) based on vegetative features [5], its reproductive morphology and present molecular phylogenetic analysis placed it as sister to *M. ornata* and *M. velutina*.

The six tepals fused as a perianth tube and actinomorphic flower of *M. nanensis* is a reverse evolution in Musaceae. Moreover, among angiosperm lineages, bilateral floral symmetry has evolved multiple times from the radial symmetric ancestors in response to natural selection associates with adaptations to pollinators [49,50,51].

### 3.1. Identification of Musa Species Using Floral Morphological Characters

In our present banana phylogenetic trees, *M. acuminata*, *M. serpentina*, *M. rubra* (=*M. laterita*), *M. siamensis* and *M. rosea* are not separated. However, these species are different from each other based on morphological features, i.e., short elliptic free tepal with long apex wing found in *M. serpentina*, *M. rubra* (=*M. laterita*) and *M. siamensis* and long ovate free tepal found in *M. rosea* as seen in Figure 8B.

*Musa rubra*, *M. laterita* and *M. siamensis* are three ambiguous species with similar morphology including itinerant rhizomes and smooth surface on subglobose seeds [52,53,54,55]. While *M. laterita* was recently reduced into a synonym of *M. rubra* [49], *M. siamensis* was reduced to a variety of *M. rubra* [56,57]. The cases were supported by the resemblance of the male flowers in all accessions of the three taxa (Figure 6).

The flowers of different subspecies of *M. acuminata* are similar, i.e., with short elliptic free tepal and short apex wing (Figure 6, Figure 7, Figure 8 and Figure 9). The result agrees with the Musaceae phylogenetic tree based on nrDNA and plastid combined data (Appendix A). Molecular phylogenetic study revealed that *M. acuminata* subspecific classification is paraphyletic [7] and the raise of each subspecies into species is not supported.

The taxonomic status of *M. flaviflora* have been unclear and it was confused with *M. thomsonii*. In 2014, Häkkinen separated *M. flaviflora* and *M. thomsonii* from *M. acuminata* subsp. *burmannica* using morphological characters such as plant waxiness, bract color and bract imbrication, while in fact, these features are variously observed in different subspecies of *M. acuminata* [58]. Moreover, we found in our investigation that floral morphological characters of *M. flaviflora* is similar to those of *M. thomsonii and M. acuminata*. This finding agreed with that of Joe et al. [59]. Therefore, we suggest that both names are synonym of *M. acuminata*. However, due to limited specimen of only flowers, subspecies cannot be defined.

### 3.2. Taxonomic Treatment of Musaceae Based on Floral Morphology



**Key to the genera of Musaceae**
1a. Outer tepal lobe apex round, lateral inner tepal acicular, style short (less than 18 mm length) ………………………………………………………………………………………… *Ensete*1b. Outer tepal lobe apex acute, lateral inner tepal ovate in shape, style long (more than 18 mm length) …………….…………………………………………………………………………… 22a. Stigma clavate or spatulate ………………..……………………………………………. *Musa*2b. Stigma capitate …………..………………………………………………… *Musella lasiocarpa*
**Key to species of *Ensete***
1a. Length/width ratio of fused tepal less than 3:1, base of median inner tepal ob-tuse……………………………………………………………………………………...... *E. homblei*1b. Length/width ratio of fused tepal more than 3:1, base of median tepal acute or subcor-date…………………………………………………………………………………………………. 22a. Base of median inner tepal acute, margin of median inner tepal shoulder den-tate……………………………………………………………………………………….... *E. glaucum*2b. Base of median inner tepal subcordate, margin of median inner tepal shoulder entire..33a. Shoulder of median inner tepal round ………………………………….…………. *E. gilletii*3b. Shoulder of median inner tepal acute………………………………………...... *E. superbum*
**Key to the section of *Musa***
1a. Median inner tepal ovate or elliptic in shape, or missing, stigma clavate …………..………………………………………………….. sect. *Musa* (including *Rhodochlamys*1b. Median inner tepal oval-lanceolate, oblanceolate, oblong or obovate, stigma spatulate ……………………………………….... sect. *Callimusa* (including *Australimusa* & *Ingentimusa*)
**Key to species of the section *Musa***
1a. Median inner tepal ovate, shoulder margin repand ……………………………………… 21b. Median inner tepal elliptic, shoulder margin dentate or entire, or fused with lateral outer tepal ………………………………………………………………………………………….. 32a. Perianth ventricose, anther ensiform …………………………………………. *M. nagensium*2b. Perianth tubular, anther oblong……………………………………………………………… 43a. Flowers nearly actinomorphic; median inner tepal fused with lateral outer tepals at adaxial side; fertile stamens six, filaments basally united, anther ensiform…… *M. nanensis*3b. Flower zygomorphic; median inner tepal elliptic and free; fertile stamens five, free, an-ther oblong………………………………………………………………………………………….. 54a. Wart-like structures on styles near stigma absent ……………………………… *M. velutina*4b. Wart-like structures on styles near stigma present ………………………………. *M. ornata*5a. Wrinkle on median inner tepal absent; surface of stigma velvet……………. *M. balbisiana*5b. Wrinkle on median inner tepal present; surface of stigma smooth …………………….. 66a. Perianth tubular; length/width ratio of fused tepal less than 3:1…. *M. rubra*, *M*. *siamensis*6b. Perianth ventricose; length/width ratio of fused tepal more than 3:1 ………………….. 77a. Spine-like dorsal appendage on lateral inner tepal lobes present ……………*M. itinerans*7b. Spine-like dorsal appendage on lateral inner tepal lobes absent ……………………….. 88a. Apex of median inner tepal truncate……………………………………………. *M. serpentina*8b. Apex of median inner tepal acute ………………………………………………………....... 99a. Median inner tepal base truncate, wing on median inner tepals present………………... …………………………………………………………………………………….….*M. yunnanensis*9b. Median inner tepal base obtuse, wing on median inner tepals absent…………………... ……………………………………………………………………………*M. acuminata*, *M. flaviflora*
**Key to the species of the section *Callimusa***
1a. Fused tepal length to median inner tepal length ratio 1–2 …………….………………… 21b. Fused tepal length to median inner tepal length ratio 3–4 ………………………………. 32a. Spine-like dorsal appendage on median outer tepal lobes absent, median inner tepal apex acuminate, wing on shoulder of median inner tepal present and margin on shoulder of median inner tepal dentate ………………………………………………………... *M. gracilis*2b. Spine-like dorsal appendage on median outer tepal lobes present, median inner tepal apex subobtuse, wing on shoulder of median inner tepal absent and margin on shoulder of median inner tepal entire ………………………………………………………….………… 43a. Median inner tepal obovate, margin of median inner tepal shoulder dentate, shoulder of median inner tepal acute, wrinkle on median inner tepal present ….………... *M. textilis*3b. Median inner tepal oval-lanceolate or oblanceolate, margin of median inner tepal shoulder entire, shoulder of median inner tepal round, wrinkle on median inner tepal ab-sent ………………………………………………………………………………………………… 54a. Spine-like dorsal appendage on lateral inner tepal lobes present, shoulder of median inner tepal round and ovary transverse section angular…………………… *M. paracoccinea*4b. Spine-like dorsal appendage on lateral inner tepal lobes absent, shoulder of median inner tepal truncate and ovary transverse section cylindrical ……………………………... 65a. Median inner tepal oval-lanceolate and median inner tepal base obtuse……*M. beccarii*5b. Median inner tepal oblanceolate and median inner tepal base acute ….……..…………… ………………………………………………………………………………… *M. maclayi*, *M. ingens*6a. Spine-like dorsal appendage on lateral outer tepal lobes absent…………... *M. haekkinenii*6b. Spine-like dorsal appendage on lateral outer tepal lobes present…………..… *M. coccinea*


## 4. Materials and Methods

### 4.1. Taxon Sampling

Fifty-nine banana accessions in 21 taxonomic categories (Appendix A) from three genera were included in this study. Most of the banana specimens used were collected from natural habitats (31 accessions) and others from cultivated areas in Thailand and China (18 and 2 accessions, respectively) during 2005 and 2017. Voucher specimens were deposited at Suan Luang Rama IX (SL) and Forest (BKF) herbaria in Bangkok, Thailand. The rest of the samples in our analysis eight accessions were dry floral specimens generously provided by Royal Botanic Gardens Kew (K) and Royal Botanic Gardens Edinburgh (E) herbaria. For morphological studies, freshly collected mature flowers were kept in 70% ethanol and dried specimens were soaked in hot water and kept in 70% ethanol before being observed.

### 4.2. Molecular Methods

Genomic DNA was isolated from flesh leaves using modified method from CTAB [60]. Four regions including the internal transcribed spacers of the nuclear ribosomal DNA (nrDNA, ITS1 + 5.8S + ITS2), *trnL* intron, 3′ *trnL* exon and intergenic spacer region (*trnL*-*F*), *rps16* intron and *atpB-rbcL* spacer were amplified. The primers of these regions were generated including ITS4 and ITS5 of ITS [61], Lc and Ff of *trnL-F* [62], rpsMF and rpsMR2 of *rps16* intron [63] and BO1 and BO2 of *atpB-rbcL* [64]. Genomic DNA of 50–200 ng were used in 25-µL PCR reactions with Vivantis taq (manufacturer, Malaysia), which contained 0.5 pmol each of forward and reverse primers, 0.4 U DNA polymerase, 4 µmol dNTPs, 2 µg buffer and 2.5 mmol/L MgCl2. The PCR conditions were as follows: initial polymerase activation at 94 °C for 2 min, 40 cycles of 94 °C for 30 s, 50–62 °C for 60 s, and 72 °C for 30 s, with a final elongation at 72 °C for 7 min. Amplicon size was verified by 1% (*w/v*) agarose gel electrophoresis. A band of appropriate sizes was purified using GEL/PCR Purification Mini Kit (Favorgen Biotech Corporation, Ping-Tung, Taiwan). PCR products were sent to Bioscience, Korea for sequencing. The sequences of Musaceae and the other families within Zingiberales were obtained including 216 sequences from Genbank and 10 new sequences that have been deposited under accession number in Appendix A.

### 4.3. Sequence Alignment and Phylogeny Reconstruction

The sequences were aligned in MAFFT [65] and edited manually using Bioedit v7.0.5 [66]. Substitution rates for Bayesian analyses were selected under the Akaike information criterion (AIC) using Jmodeltest2 on XSEDE (2.1.6) [67] and the Bayesian inference (BI) models performed was conducted using MrBayes 3.2.7a [68], both on the CIPRES web portal [69]. Substitution model was selected as GTR+I+G for ITS and GTR+G for *rps16*, *trnL-F* and *atpB-rbcL* with number of substitutions as six. The Markov Chain Monte Carlo (MCMC) was performed using independent runs with four chains, 10,000 print results, saving ten every 1000 generations, for a total of five million of generations.

Maximum Likelihood was reconstructed using RAxML [70] on CIPRES portal with GTRGAMMAI model and four chains to parallel search the tree. A total of 1000 bootstrap replicates were generated by random sequence addition. To generated a starting tree for the parsimony inference, 12,345 was a random number used.

### 4.4. Multivariate Analysis

Fifteen quantitative characters (Figure 10) of the floral structures including tepals, stamens and pistils of ten male flowers (three flowers from each herbarium specimen, except for a scarce specimen which only one flower was used) from either fresh or dried accessions were measured using ruler (Appendix A). The range of variation between taxa on each quantitative character was visualized asbox-plots and transferred into decimal logarithms. Correlations between variations of quantitative data were determined by PCA. To estimate homogeneity of the morphology within groups of Musaceae, CDA was applied. Stepwise discriminant analysis, unstandardized coefficients and Maholanobis distance were used to determine characters that separate groups. Principal component and discriminant scores were constructed into scatter plots to identify groups within Musaceae. PCA and CDA were computed using PASW Statistics 18 (SPSS, Inc.). To determine similarities between samples, Gower similarity index was computed and the distributions of quantitative variables were visualized as box-plot and violin-plot in Past 4.05 [71].

### 4.5. Morphological Analyses

Ten male flowers (one to three flowers from each herbarium specimens) were observed under dissecting microscope (Olympus SZ40 light microscope, Tokyo, Japan). Twenty-two qualitative characters of the floral structure including two whorls of tepal, which composed of three outer and three inner tepals, stamen and pistil were coded into binary or multistate variables. To determine similarities between samples, Gower similarity index was perform using Past 4.05 [71]. The UPGMA performed on 59 OTUs was based on 22 floral morphological characters, of which states were coded as in Table 5. These characters were used to construct an identification key to the banana species in this present study.

### 4.6. Character Evolution and Analysis

Using fresh flowers, dried specimens or pictures from original publication, five qualitative characters of male flowers including perianth symmetry (bilateral, radial and asymmetric), lateral inner tepal shape (acicular and ovate), median inner tepal shape (tricuspidate, obovate, oval-lanceolate, oblanceolate, oblong, elliptic, ovate and fused with perianth tube), stigma shape (clavate, spatulate and capitate) and style length (shorter than 18 mm and longer than 18 mm) were investigated (Appendix A). The data matrix was coded into binary or multistate variables with unordered states. The species of which specimens cannot be observed were treated as a missing data. Evolution of the floral characters was traced over the phylogenetic trees using Mesquite version 3.6 [72]. The selected characters were constructed into identification keys to genera and *Musa* sections. We used Fitch parsimony [73] as a criterion for character optimization. To account for phylogenetic uncertainty, we traced character histories on 5001 post burn-in trees from the Bayesian analysis using the ‘Trace Character Over Trees’ command in Mesquite 3.6 [71].

## 5. Conclusions

The morphological characters of the male flowers within the banana family (Musaceae) show several distinguishable characters for the classification. These characters were used to construct a key to genera and sections in the genus *Musa*. The cluster analysis using PCA and CDA indicated that the three genera, *Ensete*, *Musa* and *Musella*, are different with supports by degrees of perianth fusion, median inner tepal shapes, lengths of styles and stigma shapes. Molecular phylogenetic analysis of ITS, *trnL-F*, *rps16* and *atpB-rbcL* indicated that *Musella* is closer related to *Ensete* than to *Musa*. Based on capitate stigma, supporting by anther vegetative and reproductive morphological characters, *Musella* should be treated as a distinct genus, and not as a member of the genus *Ensete*. The elliptic and ovate shape of the median inner tepal and capitate stigma can clearly be used to separate the section *Musa* (including *Rhodochlamys*) from the section *Callimusa* (including *Australimusa* and *Ingentimusa*) that have obovate, oval-lanceolate, oblanceolate and oblong shapes combine with spatulate stigma. The species with undetermined section, *M. nanensis* with its uniquely different flowers including radial symmetry, six stamens and their fused filaments, should be placed in the section *Musa* based on its molecular phylogeny and the occurrence of a clavate stigma. The results and analyses from this study provide significant information on male floral characters as key characters to classify genera and section within Musaceae.

## Figures and Tables

**Figure 1 plants-12-01602-f001:**
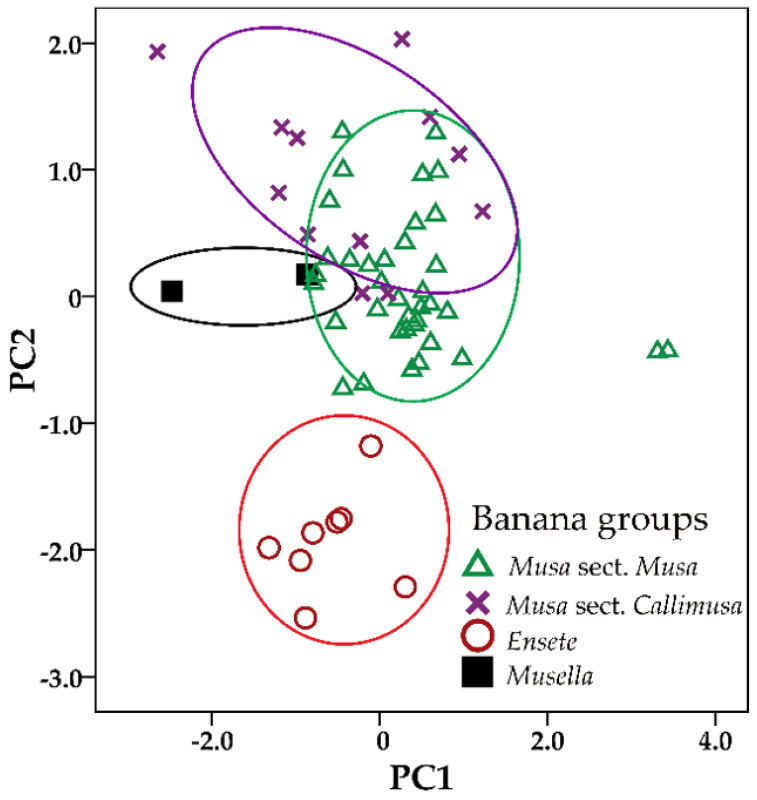
Scatter plot of the first two principal components based on 15 male flower traits from 59 OTUs of Musaceae. Symbol representative group of Musaceae: triangle, *Musa* + *Rhodochlamys*; cross, *Callimusa* + *Australimusa*; circle, *Ensete* and filled square, *Musella*.

**Figure 2 plants-12-01602-f002:**
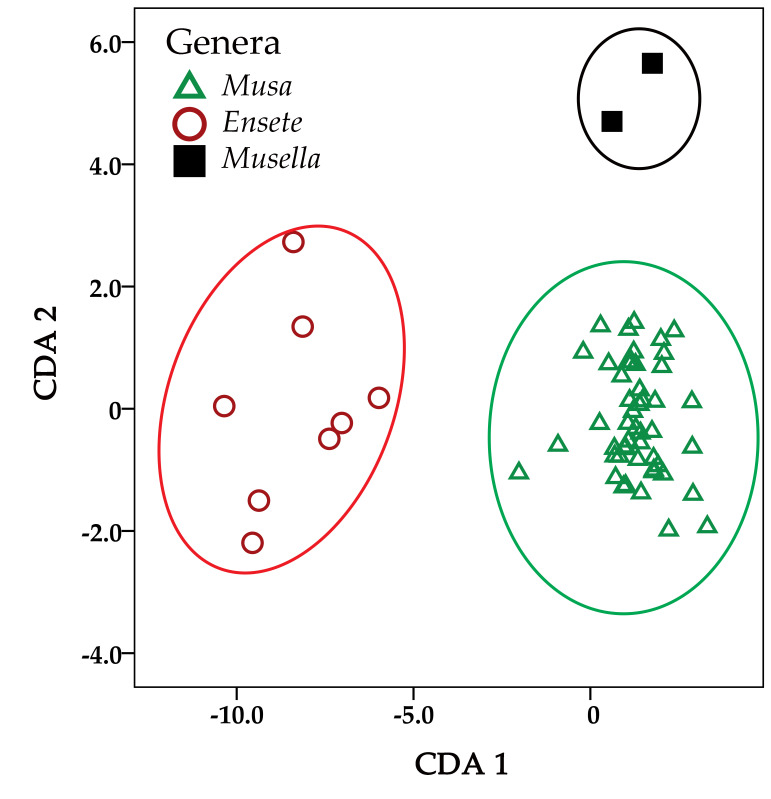
Scatter plot of the first two discriminant axis based on 15 male flower traits from 59 OTUs shows grouping of *Musa*, *Ensete* and *Musella* within the Musaceae.

**Figure 3 plants-12-01602-f003:**
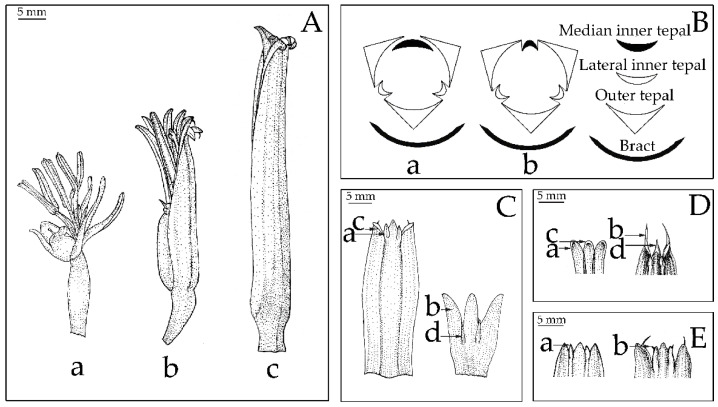
Compound tepal morphologies comparison and character state between Musaceae (character no. 1–8), (**A**) shape of perianth: (**a**) fused tepal with deep lobes; (**b**) ventricose; (**c**) tubular, (**B**) symmetry of flower: (**a**) bilateral symmetry; (**b**) radial symmetry, (**C**) shape of tepal lobe: (**a**) acute outer tepal apex lobes; (**b**) round outer tepal apex lobes; (**c**) ovate lateral inner tepal lobes; (**d**) acicular lateral inner tepal lobes, (**D**) spine-like dorsal appendage on outer tepal lobe: (**a**) absent on lateral outer tepal lobes; (**b**) present on lateral outer tepal lobes; (**c**) absent on median outer tepal lobes; (**d**) present on median outer tepal lobes, (**E**) spine-like dorsal appendage on lateral inner tepal lobes: (**a**) absent; (**b**) present.

**Figure 4 plants-12-01602-f004:**
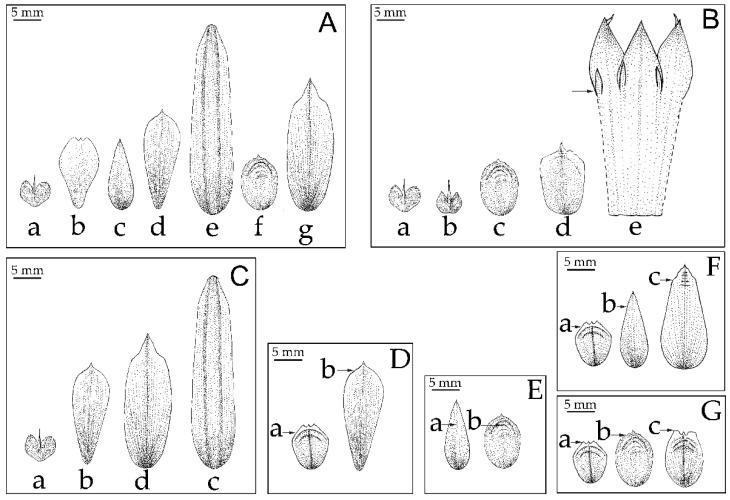
Median inner tepal morphologies comparison and character state between Musaceae (character no. 9–15), (**A**) shape of free median inner tepal: (**a**) tricuspidate; (**b**) obovate; (**c**) oval-lanceolate; (**d**) oblanceolate; (**e**) oblong; (**f**) elliptic; (**g**) ovate, (**B**) base of median inner tepal: (**a**) acute; (**b**) sub-cordate; (**c**) obtuse; (**d**) truncate; (**e**) adnate with perianth tube (**C**) apex of free median inner tepal: (**a**) cuspidate; (**b**) acuminate; (**c**) attenuate; (**a**) subobtuse (**D**) wings on shoulder of free median inner tepal: (**a**) wing present; (**b**) wing absent, (**E**) present of wrinkle on free median inner tepal: (**a**) absent; (**b**) present, (**F**) margin on shoulder of free median inner tepal: (**a**) dentate; (**b**) entire; (**c**) repand, (**G**) apex of shoulder of free tepal: (**a**) acute; (**b**) round; (**c**) truncate.

**Figure 5 plants-12-01602-f005:**
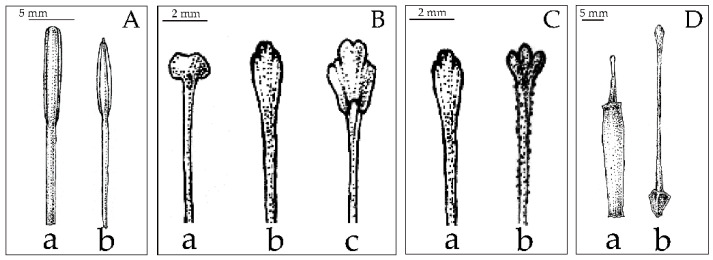
Stamen and pistil morphologies comparison and character state between Musaceae (character no. 17–21), (**A**): shape of anther: (**a**) oblong; (**b**) ensiform, (**B**) shape of stigma: (**a**) capitate; (**b**) clavate; (**c**) spatulate, (**C**) wart-like structure present on style surface: (**a**) absent; (**b**) present, (**D**)shape: (**a**) cylindrical; (**b**) angular.

**Figure 6 plants-12-01602-f006:**
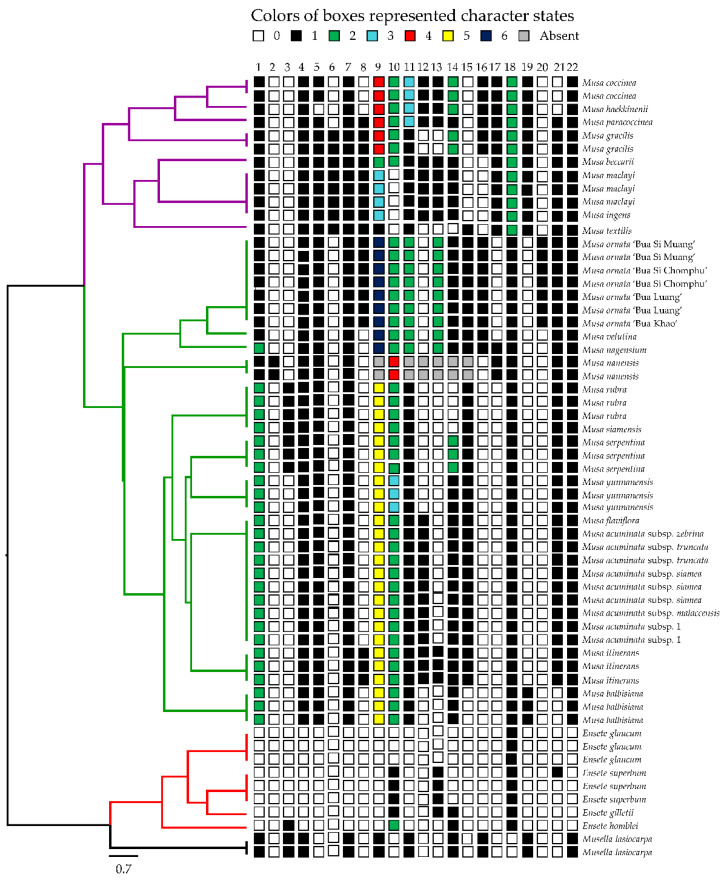
UPGMA dendrogram of 59 OTUs within Musaceae based on 22 floral morphological characters of male flowers (see details of the character states in Materials and Methods, character no. 1–8 in Figure 3; character no. 9–15 in Figure 4; character no. 17–21 in Figure 5; Table 5).

**Figure 7 plants-12-01602-f007:**
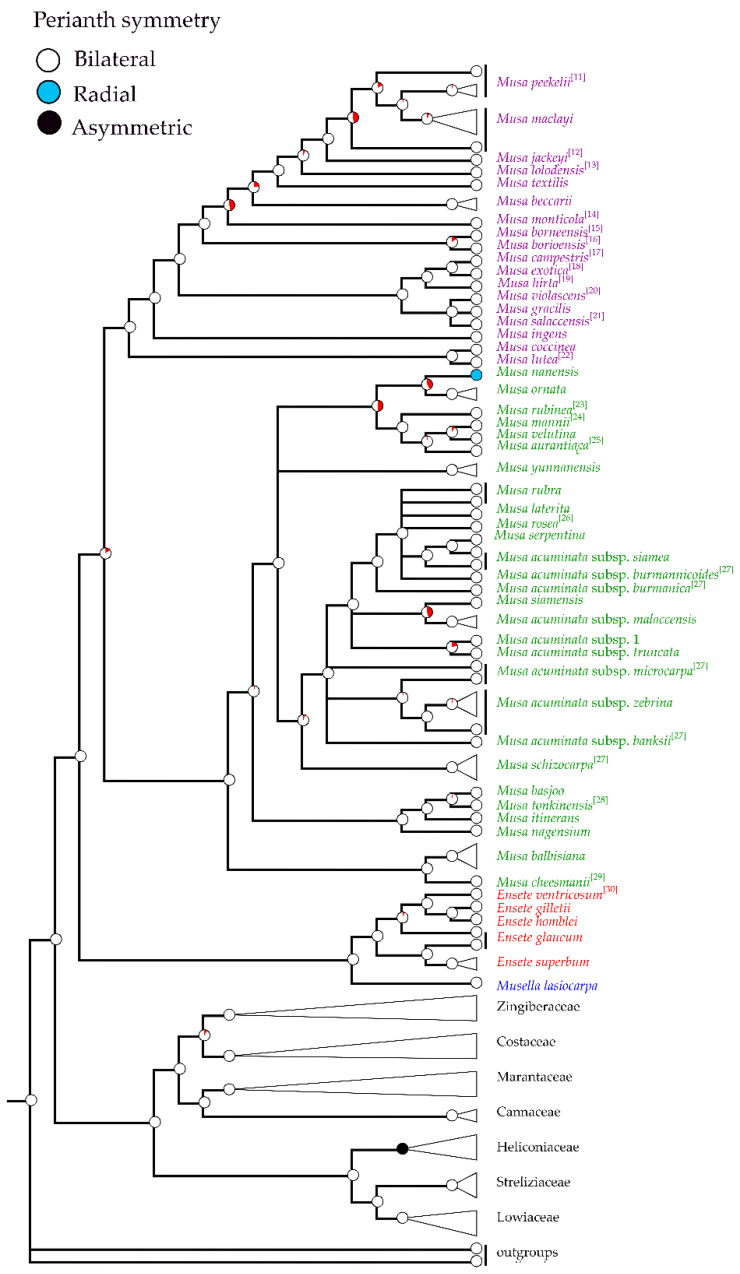
Ancestral state reconstruction, based on Bayesian posterior probabilities of selected morphological states on Bayesian topology constructed from nrDNA and plastid combined data. The tree shows the evolution of the perianth symmetry in the order Zingiberales with emphasis on Musaceae. The phylogenetic relationship included the families within Zingiberales (Musaceae, Strelitziaceae, Lowiaceae, Heliconiaceae, Costaceae, Cannaceae, Marantaceae and Zingiberaceae). The tree separated banana clades including *Musella* (blue), *Ensete* (red), species of sect. *Musa* (green) and species of sect. *Callimusa* (purple). Circles on nodes show character changes between the taxa, red pie represent node absent. Morphological characters from other publications were labeled with reference number [11,12,13,14,15,16,17,18,19,20,21,22,23,24,25,26,27,28,29,30].

**Figure 8 plants-12-01602-f008:**
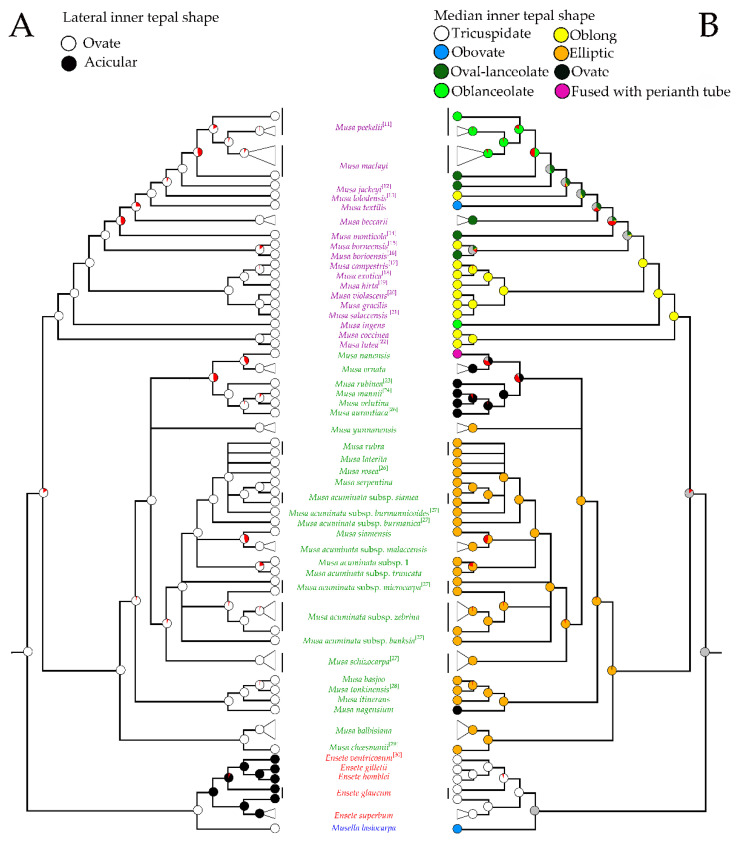
Phylogenetic trees of Musaceae based on the lateral inner tepal shapes (**A**) and the median inner tepal shapes (**B**) of male flower. The trees show ancestral state reconstruction based on Bayesian posterior probabilities of selected morphological states on Bayesian topology constructed from nrDNA and plastid combined data. The tree separated banana clades including *Musella* (blue font), *Ensete* (red font), *Musa* species of sect. *Musa* (green font) and *Musa* species of sect. *Callimusa* (purple font). Circles on nodes show character changes between the taxa, red pies represent node absent and gray pie represent equivocal. Morphological characters from other publications were labeled with reference number [11,12,13,14,15,16,17,18,19,20,21,22,23,24,25,26,27,28,29,30].

**Figure 9 plants-12-01602-f009:**
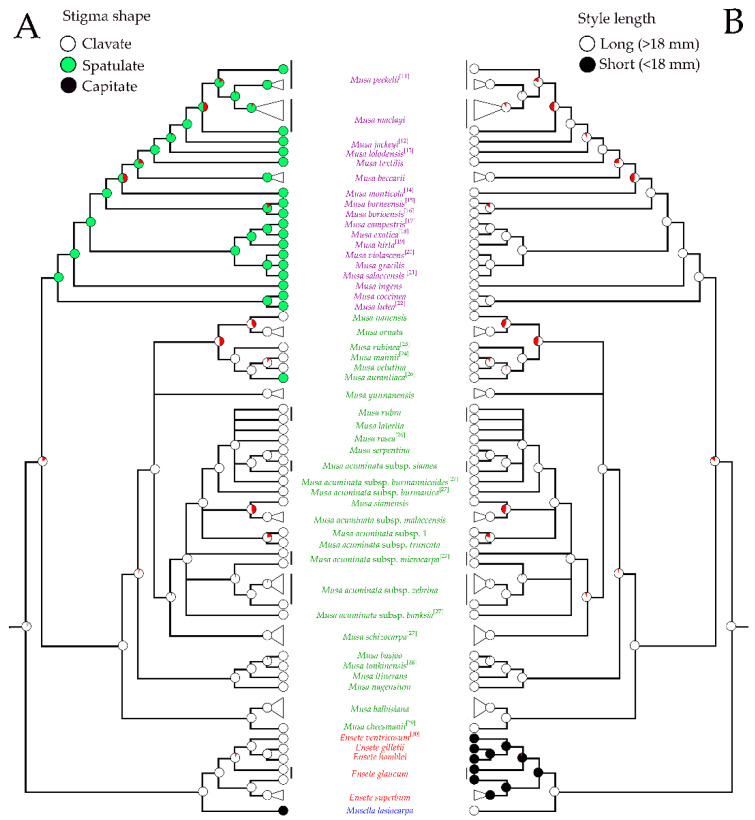
Phylogenetic trees of Musaceae based on the stigma shapes (**A**) and the style lengths (**B**) of male flowers. The trees show ancestral state reconstruction based on Bayesian posterior probabilities of selected morphological states on Bayesian topology constructed from nrDNA and plastid combined data. The tree separated banana clades including *Musella* (blue font), *Ensete* (red font), species of the sect. *Musa* (green font) and species of the sect. *Callimusa* (purple font). Circles on nodes show character changes between the taxa, red pies represent node absent and gray pie represent equivocal. Morphological characters from other publications were labeled with reference number [11,12,13,14,15,16,17,18,19,20,21,22,23,24,25,26,27,28,29,30].

**Figure 10 plants-12-01602-f010:**
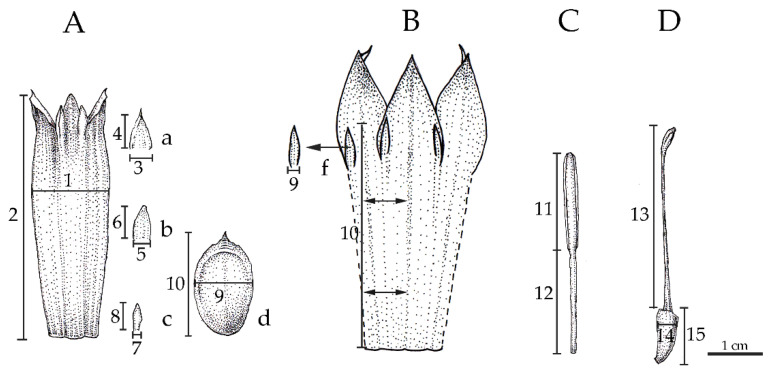
Quantitative characters of banana flowers. (**A**), Perianth of bilateral flower of banana including (**a**) lateral outer tepal, (**b**) median outer tepal, (**c**) later inner tepal and (**d**) median inner tepal (free tepal) of all Musaceae except *M. nanensis*, (**B**), perianth of *M. nanensis* that (**f**) median inner tepal fused with perianth tube, (**C**), stamen and (**D**,) pistil, i.e., 1. width of fused tepal, 2. length of fused tepal, 3. width of lateral outer tepal lobe, 4. length of median outer tepal lobe, 5. width of central outer tepal lobe, 6. length of central outer tepal lobe, 7. width of inner tepal lobe, 8. length of inner tepal, 9. width of median inner tepal, 10. length of median inner tepal, 11. length of anther, 12. length of filament, 13. length from style base to stigma head, 14. width of ovary, 15. length of ovary. Scale bar = 1 cm.

**Table 1 plants-12-01602-t001:** Principal component scores between different groups within Musaceae. The table shows corresponding variation from the first four components with eigenvalues greater than 1 of 15 quantitative male floral morphological characters of 59 OTUs.

Component	Initial Eigenvalues		
	Total	% of Variance	Cumulative %
1	5.289	35.26	35.26
2	3.980	26.53	61.79
3	1.496	9.98	71.76
4	1.312	8.75	80.51

**Table 2 plants-12-01602-t002:** Variable loadings in principal component analysis from 15 male floral morphological characters of *Musa*, *Ensete* and *Musella*. Loading higher than 0.7 are marked with bold type face.

Component Matrix	Component			
	1	2	3	4
1. width of compound tepal	0.730	0.345	−0.199	−0.297
2. length of compound tepal	0.675	0.518	0.292	0.193
3. width of lateral outer compound tepal lobe	0.845	−0.267	−0.296	−0.173
4. length of lateral outer compound tepal lobe	0.808	−0.465	−0.094	−0.010
5. width of central outer compound tepal lobe	0.861	−0.332	−0.161	0.008
6. length of central outer compound tepal lobe	0.756	−0.567	−0.090	0.036
7. width of inner compound tepal lobe	0.601	0.372	−0.365	0.353
8. length of inner compound tepal lobe	0.261	−0.829	0.304	−0.101
9. width of median inner tepal	−0.274	0.187	0.426	0.281
10. length of median inner tepal	0.280	0.824	0.157	−0.068
11. length of anther	0.609	−0.017	0.432	0.538
12. length of filament	0.389	0.597	0.265	−0.499
13. length from style base to stigma head	0.475	0.733	−0.105	0.375
14. width of ovary	0.487	0.206	0.539	−0.478
15. length of ovary	0.227	−0.630	0.502	0.169

**Table 3 plants-12-01602-t003:** Summary of canonical discriminant function of four clustering groupings of 59 OTUs between *Musa*, *Ensete* and *Musella* within Musaceae showed percent of variance in analysis.

Eigenvalues					
Function	Eigenvalue	% of Variance	Cumulative %	Canonical Correlation	
1	11.316	91.9	91.9		0.959
2	0.996	8.1	100		0.706

**Table 4 plants-12-01602-t004:** Pooled within-groups correlations between discriminating variables and standardized canonical discriminant functions of Musaceae. Variables ordered by absolute size of correlation within function. The correlations of floral characters which most significantly distinguished *Ensete* from *Musa* and *Musella* are indicated in bold typeface in Function 1, meanwhile, those that separated *Musella* from *Ensete* and *Musa* are in bold typeface in Function 2.

Structure Matrix	Function	
	1	2
Length from style base to stigma head	0.662	−0.461
Width of ovary	0.252	−0.064
Length of median inner tepal	0.231	−0.031
Length of filament	0.181	−0.169
Length of ovary	−0.135	0.068
Length of compound tepal	0.200	−0.683
Length of anther	0.024	−0.487
Width of lateral outer compound tepal lobe	−0.081	−0.394
Width of compound tepal	−0.069	−0.393
Length of inner compound tepal lobe	−0.109	−0.389
Width of central outer compound tepal lobe	−0.064	−0.353
Length of lateral outer compound tepal lobe	−0.083	−0.241
Width of inner compound tepal lobe	0.001	−0.195
Length of central outer compound tepal lobe	−0.134	−0.160
Width of median inner tepal	0.021	0.105

**Table 5 plants-12-01602-t005:** The 22 male floral morphological characters and their states used in the analysis.

No.	Male Floral Morphological Characters	Character States
1	Shape of perianth	(0) fused tepal with deep lobes
		(1) ventricose
		(2) tubular
2	Symmetry of perianth	(0) bilateral symmetry
		(1) radial symmetry
3	Ratio of length/width of fused tepal	(0) 3–4
		(1) 1–2
4	Shape of outer tepal apex lobe	(0) round
		(1) acute
5	Spine-like dorsal appendage present on lateral outer tepal lobe	(0) absent
		(1) present
6	Spine-like dorsal appendage present on median outer tepal lobe	(0) absent
		(1) present
7	Shape of lateral inner tepal lobe	(0) acicular
		(1) ovate
8	Spine-like dorsal appendage present on lateral inner tepal lobe	(0) absent
		(1) present
9	Shape of median inner tepal	(0) tricuspidate
		(1) obovate
		(2) oval-lanceolate
		(3) oblanceolate
		(4) oblong
		(5) elliptic
		(6) ovate
10	Base of median inner tepal	(0) acute
		(1) sub-cordate
		(2) obtuse
		(3) truncate
		(4) adnate with fused tepal
11	Apex of free median inner tepal	(0) cuspidate
		(1) acuminate
		(2) attenuate
		(3) subobtuse
12	Wings on shoulder of free median inner tepal	(0) present
		(1) absent
13	Margin of free median inner tepal shoulder	(0) dentate
		(1) entire
		(2) repand
14	Shoulder of free median inner tepal	(0) acute
		(1) round
		(2) truncate
15	Present of wrinkle on free median inner tepal	(0) absent
		(1) present
16	Ratio of median inner tepal length/fused tepal length	(0) 1/3–1/4
		(1) 1/1–1/2
17	Shape of anther	(0) oblong
		(1) ensiform
18	Shape of stigma	(0) capitate
		(1) clavate
		(2) spatulate
19	Surface of stigma	(0) smooth
		(1) velvet
20	Wart-like structure present on style surface	(0) absent
		(1) present
21	Ovary transvers shape	(0) cylindrical
		(1) angular
22	Style length	(0) less than 18 mm
		(1) more than 20 mm

## Data Availability

Not applicable.

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
