# Peer review of "Evolution and Classification of Musaceae Based on Male Floral Morphology"

_plants, 2023, doi:10.3390/plants12081602_

Round 1
Reviewer 1 Report
This paper provides a relatively comprehensive classification of 59 banana accessions of 21 taxa using molecular marker identification and comparison of morphological indicators. Additionally, information on male flower traits of Musaaceae is also described, which provides an important reference for subsequent research.
The findings are clear, however, there are still three concerns that need to be addressed before this manuscript can be considered acceptable.
1.Although the nrDNA and plastid gene fragments used in molecular evidence can accurately identify the accessions, it would be interesting to explore the possibility of using MADS-box related genes to resolve disputed species.
2.The chapter title is confusing, and part 2.2 is missing, while part 2.1 is repeated. Please carefully review and modify the chapter title.
3.Some diagrams are missing the ruler.
Author Response
Dear reviewer,
Thank you for your comments and suggestions. Here are our answers to your inquiry.
- The MAD-box related genes are very interesting for identification of the banana species. It could be included in our on-going study on gene structure and expression to explained the difference of floral symmetry and morphology.
- Thanks to you, the title name has been changed to “Evolution and Classification of Musaceae based on Male Floral Morphology.” We also modify the chapter titles as following; 2.2 Multivariate analysis, 2.4 Morphological analysis.
- Wandee has added the rulers in the graphical figures.
With best regards,
Sasivimon & Wandee

Reviewer 2 Report
The revieved paper is devoted to the phylogeny of Musaceae together with associated evolution and taxonomic significance of some reproductive features.
I have numerous minor concerns about this work which can be found in a manuscript file (see attached). The overall style and language of this paper require a deep elaboration. Although I have made some corrections, much more are needed, as in some parts it is difficult to understand the exact meaning of a phrase. Authors are recommended to have their manuscript checked by either native speaker or specialized editing service.
It is not clear how many flowers of each species were examined, as there are contradictory statements in this text. This point needs to be specified.
Figure 1 can be probably elaborated to remove most of non-Musaceae Zingiberales. At the moment, they comprise ca. 2/3 of the whole tree which seems excessive for the understanding of relations within Musaceae.
I also suggest to elaborate the Discussion section, as references to pollination syndromes seem somewhat unfounded at the moment. The minor differences in tepal shape or style length can hardly be decisive for the pollination syndrome.
After these suggestions are considered, this manuscript can be reconsidered again to be recommended for publication in Plants.

Author Response
Dear reviewer,
We wish to thank you for suggestions and comments. Changes we have made in our revised version are as following.
- We have renamed the title into “Evolution and Classification of Musaceae based on Male Floral Morphology in Musaceae” to better represent main parts of our research as your suggestion.
- The Figure 1 of the Bayesian tree has been moved to supplementary file. The tree is specifically useful to trace direction of evolution within Musaceae, but the floral morphology of other families within Zingiberales was not observed and the tree can be omitted from the main result part. However, we left it in the supplementary part because we think it can be of some uses to other Zingiberales researchers in the future.
- We have deleted the Figure 4 (drawings of floral morphological variations) from the article because we have realized that the drawings already present in the Figure 10,11 &12.
- In Figure 6, 7 & 8, we added reference numbers at the taxa names as your suggestion.
- In Figure 8, we changed some character-state box colour for better visualization.
- We added rulers in the Figure 10, 11 & 12.
- In the section 4.5 Morphological Analysis, we put 22 floral characters into the Table 5.
- We have checked reference format and rewritten them.
- The gramma errors were edited according to the suggestions. Wherever we do not agree with the suggestions, we did explain.
With best regards,
Sasivimon & Wandee

Reviewer 3 Report
Review of the article: Evolution of Male Floral Morphology in Musaceae
Summary
The article begins with a phylogenetic analysis based on DNA sequence of the fam. Musaceae in which species of the genus Musa, Ensete and Musella result clearly separated. It is followed by the PCA of 15 quantitative characters of floral morphology, resulting in Ensete separated from Musa and Musella. Male floral characters are presented as key characters for the classification of genera in the Musaceae.
General comment
The article contains an interesting combination of morphological and molecular data. The statistical treatment is correct and also the presentation of the results. Nevertheless, it requires some corrections as indicated below for each of the sections.
Comments by sections
Abstract
Please correct:
Phylogenetic trees of Musaceae and the other family within the Zingiberales were reconstructed from 226 sequences of…
To:
Phylogenetic trees of Musaceae and the other families within the Zingiberales were reconstructed from 226 sequences of…
Results
Arrange the numbering of sections:
2.1. Phylogenetic analysis
2.1. Multivariate análisis (on p. 4)
2.1. Multivariate análisis (on p.11)
Fig. 1 is too large and the species names are not visible. I suggest to divide it in two parts Leaving room enough to read all the names of the species analysed, in particular in the Musaceae.
Rows 113-115:
triangle; Musa plus Rhodochlamys, plus; Callimusa plus Australimusa, circular;
Ensete, filled square;
Better:
Triangle: Musa and Rhodochlamys; +: Callimusa and Australimusa; circle:
Ensete; filled square:Musella.
2.3. Morphological character in classification of Musaceae
(should be Morphological characters in classification of Musaceae)
Row 138:
However, the tepals of M. nanensis are fused into a floral tube representing an actinomorphic flower (Figure. 4A).
Instead:
However, the tepals of M. nanensis are fused into a floral tube representing an actinomorphic flower (Figure. 4Ab).
Legend to Figure 5. At the end make clear:
Circles on nodes show character changes between the taxa, red pie represent node absent and gray pie represent equivocal.
There are nodes without posterior character change with diverse degrees of red. There is no gray color in this figure, only red and white.
Legend to Figure 6. All this information is redundant. It is given in the figure:
“In A, white circles representing ovate and black circles represent acicular. In B, white circles representing a tricuspidate, blue circles representing an obovate, dark green representing oval-lanceolate, light green circles representing oblanceolate, yellow circles representing oblong, orange circles representing elliptic, black circles representing ovate and purple circles representing fused with perianth tube.”
Morphological characters from other publications were label with star (*).
(Please indicate ref. number for these publications).
Minor questions:
Row 61, 84. Correct (remove the dot):
(Figure. 1), (Figure. 2)....(In all figure mentions).
Row 69: Legend to Figure Complete the sentence
Bayesian posterior probabilities (PP) and are labelled on the nodes.
Row 185: The tree showed the evolution of perianth symmetry of the family
Change to:
The tree shows the evolution of perianth symmetry in the order Zingiberales with emphasis on the family Musaceae…
Better: The tree shows the evolution of perianth symmetry of the family
Row 186:
(M. nanensis with arrow)
Where is the arrow?
188:
The phylogenetic relationship included the family within Zingiberales
The phylogenetic relationship includes the families within Zingiberales
189:
Strelitziacea,
Strelitziaeae,
190:
beraceae. The…
Close parenthesis:
beraceae). The…
Row 391, correct:
Genomic DNA of 50-200 ng were used in 25-μl PCR reactions with Vivantis taq (manufacturer, Malaysia),
Row 397:
A band each of appropriate sizes
Row 399:
The sequences of Musaceae and the other family within Zingiberales
Change to:
The sequences of Musaceae and the other families within Zingiberales
508: Change tricupidate to tricuspidate
Author Response
Dear reviewer,
Thanks for your comments and suggestions. We have improved the manuscript and listed the revisions as following.
- We have edited the gramma error according to your kind suggestions.
- We have rearranged the numbers of sections as following: 2.1. Phylogenetic analysis, 2.2 Multivariate analysis, 2.3. Morphological characters in the classification of Musaceae, 2.4 Morphological analysis.
- 1 is too large and contains many of the taxa within order Zingiberales, therefore, Wandee decided to move the figure into the supplement files because she wants to focus only on the Musaceae family. The tree is specifically useful to trace direction of evolution within Musaceae, but the floral morphology of other families within Zingiberales was not observed and the tree can be omitted from the main result part. However, we left it in the supplementary part because we think it can be of some uses to other Zingiberales researchers in the future.
- We have removed dots from Figure. X.
- We have rewritten Figure 5 legend.
- In Figure 6, 7, 8, we have deleted redundant information from the description and we added reference numbers above the taxa on trees.
With best regards,
Sasivimon & Wandee

Reviewer 4 Report
I read the manuscript “Evolution of Male Floral Morphology in Musaceae” submitted to the journal Plants. It is one of few examples –among those I have read recently– that beautifully combines very traditional systematic work based on morphometry and qualitative traits and DNA analyses; it is clear, although detailed, well-written, and well-illustrated. A series of statistical methods seems to be performed correctly, which is a good basis for well-established conclusions and interesting discussion. I am impressed by the high quality of work done and put into this study. I would suggest publishing it with only small corrections of purely editorial character.
The list of my remarks and observations is really short:
L. 31 “Musella (Franch.) C.Y. Wu ex H.W. Li. belong to”
Delete the point after “H.W.Li”. Preferably remove spaces within the author’s abbreviations, i.e. C.Y.Wu ex H.W.Li
L. 71 “Principal component analysis”
In order of being consistent, please use Principal Component Analysis.
L. 61, 84, 89, 139, 142, 144, 147, 150, 157, 159, etc. “Figure.”
Remove the point after “Figure”
L. 331 “Curiously, Musa nanensis found recently by the senior author and her team”
What do you mean by “the senior author and her team”? Could you be more specific and re-word this fragment for better clarity?
L. 408-409 “rps16, trnL-F and atpB-rbcL”
I think these names should be written in italics.
L. 423 “Canonical discriminant analysis”
I would suggest using “Canonical Discriminant Analysis” (like Principal Component Analysis in this manuscript). Another option would be with using lowercase for all analyses.
L. 552 “DA”
Do you mean CDA?
L. 546 “Ensete Horan.”
Ensete Bruce ex Horan. according to the International Plant Names Index and Plants of the World Online? You can also delete the author as in other cases. Actually, Ensete is the only Latin name with an author in the key.
Figures 4, 10, 11 & 12.
Figures 10, 11 & 12 repeat, to some extent, drawings presented in Figure 4. I would like to suggest eliminating the superfluous ones and reorganising the figures. Please note also that the terminology used for 4D differs from 12B (spatulate and clavate vs. clavate and spatulate). In 4D, it is incorrect.
Author Response
Dear reviewer,
Thanks for your comments and suggestions. We have improved the manuscript and listed the revisions as following.
- We edited the author name, Musella (Franch.) C.Y. Wu ex H.W. Li following your suggestions.
- We edited the methods’ names, i.e., Principal Component Analysis and Canonical Discriminant Analysis and changed DA to CDA.
- We remove periods from Figures.
- The statement, “Curiously, Musa nanensis found recently by the senior author and her team..” have been changed to “Musa nanensis found recently by Swangpol and her team…”
- We have deleted the author names in the keys.
- Because the drawings in the Figure 4 can be replaced by the Figures 10, 11 & 12, we deleted Figure 4 from the manuscript.
- The Figure 12B was changed to (B) shape of stigma: (a) capitate; (b) clavate; (c) spatulate.
With best regards,
Sasivimon & Wandee

Round 2
Reviewer 1 Report
After the author's revision, the context of the manuscript becomes clearer and the logic becomes more coherent. However, there are still some detailed errors:
1: Fig6A in line 157 of the text may be information from a previous version and needs to be modified.
2: The image in line 215 needs to be added with a new image sequence number.
After correcting these errors, I will recommend accepting the publication of this study.
Author Response
Dear Reviewer,
Thank you for your kind suggestions.
1: Fig 6A in line 157 of the text may be information from a previous version and needs to be modified.
Answer: Fig 6A is now written as Fig 6.
2: The image in line 215 needs to be added with a new image sequence number.
Answer: Fig. 9, 10 , 11 are added.
Thank you
Sasivimon C. Swangpol
